

# Recent results of laser spectroscopy experiments
# of pionic helium atoms at PSI

**Masaki Hori**[1★], **H. Aghai-Khozani**[1†], **A. Sótér**[1‡], **A. Dax**[2] **and D. Barna**[3§]

**1** Max-Planck-Institut für Quantenoptik,
Hans-Kopfermann-Strasse 1, D-85748 Garching, Germany
**2** Paul Scherrer Institut, CH-5232 Villigen, Switzerland
**3** CERN CH-1211, Geneva, Switzerland

★ Masaki.Hori@mpq.mpg.de
† Current address: McKinsey and Company, Munich, Germany
‡ Current address: ETH Zürich, IPA, Zurich, Switzerland
§ Current address: Institute for Particle and Nuclear Physics,
Wigner Research Centre for Physics, Budapest, Hungary

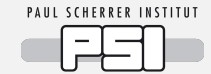

## Abstract

A review of a recent experiment carried out at PSI involving laser spectroscopy of metastable pionic helium ($\pi^4\text{He}^+ \equiv \pi^- + {}^4\text{He}^{2+} + e^-$) atoms is presented. An infrared transition $(n, \ell) = (17, 16) \rightarrow (17, 15)$ at a resonance frequency of $\nu \approx 183760$ GHz was detected.

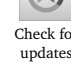
## 26.1 Introduction

Metastable pionic helium is a neutral exotic atom [1–8] that contains a helium nucleus with an electron in the ground state, and a negatively-charged pion ($\pi^-$) occupying a state having high principal and orbital angular momentum quantum numbers of around $n \sim \ell + 1 \sim 16$. These states have nanosecond-scale lifetimes against the competing cascade processes of $\pi^-$ nuclear absorption and $\pi^- \rightarrow \mu^- + \overline{\nu}_\mu$ decay. This longevity arises because the $\pi^-$ orbitals have very small overlap with the nucleus and so the rates of electromagnetic cascade processes involving the rapid deexcitation of the $\pi^-$, such as Auger and radiative decays, are significantly reduced. This characteristic recently enabled laser spectroscopy [5, 9] of $\pi^4\text{He}^+$ which constituted the first such measurement of an exotic atom that contained a meson, and showed the existence of this long-lived three-body atom. By comparing the atomic frequencies measured by laser spectroscopy with the results of quantum electrodynamics (QED) calculations, the $\pi^-$ mass [10–12] can, in principle, be determined with a high precision. This can help set upper limits on constraints on the muon antineutrino mass by laboratory experiments [13]. Some upper limits may also be set on any exotic force [14–18] that involves the $\pi^-$, as has been done in the case of antiprotonic helium ($\overline{p}\text{He}^+ \equiv \overline{p} + \text{He}^{2+} + e^-$) atoms [19–30]. Unlike the $\overline{p}\text{He}^+$ case,

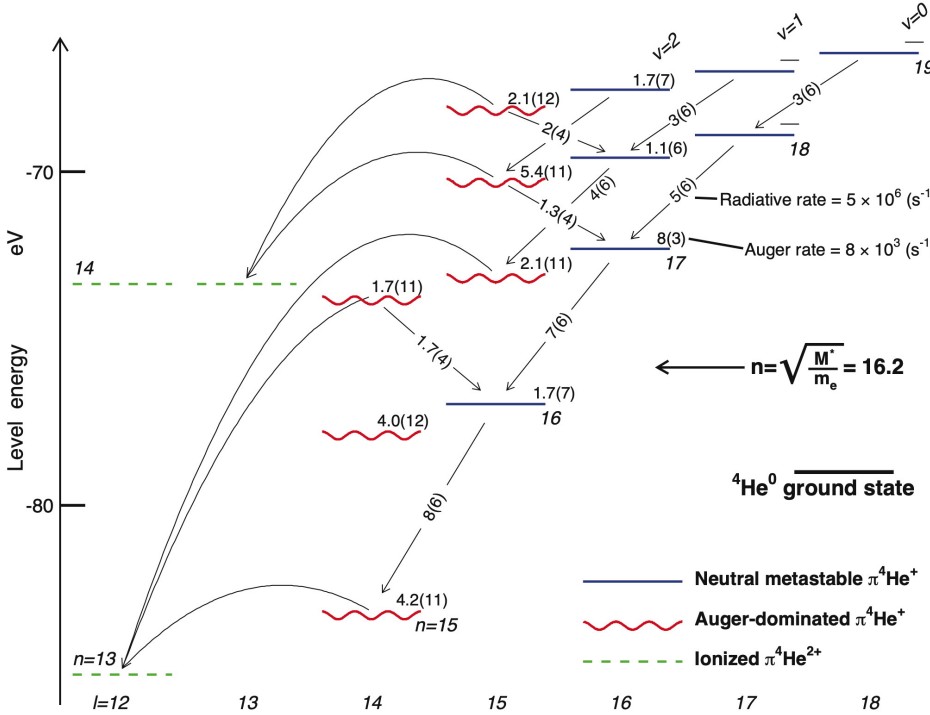

Figure 26.1: An energy level diagram of the exotic atom $\pi^4\text{He}^+$. The theoretical absolute energy of the states $(n, \ell)$ are plotted relative to the three-body-breakup threshold. The wavy lines indicate Auger-dominated states that have picosecond-scale lifetimes, and the solid lines show metastable levels with lifetimes of $> 10$ ns. The Auger decay rates are indicated in $s^{-1}$. The dashed lines show the $\pi^4\text{He}^{2+}$ ionic states which are formed after Auger electron emission. The curved arrows indicate the Auger transitions that have minimum $|\Delta \ell_A|$. The radiative transitions $(n, \ell) \rightarrow (n-1, \ell-1)$ and $(n, \ell) \rightarrow (n-1, \ell+1)$ are shown using straight arrows, with the corresponding decay rates indicated in $s^{-1}$. From [5].

the atomic structure of $\pi^4\text{He}^+$ contains no hyperfine structure that arises from the spin-spin interaction between the spin-0 $\pi^-$ and $^4\text{He}$ nucleus [31, 32].

The existence of $\pi\text{He}^+$ atoms had been inferred in an indirect way from four experiments [33–37] that were initially carried out using early synchrocyclotron facilities [38,39] and liquid helium bubble chambers [40]. All these experiments observed that some $\pi^-$ coming to rest in helium targets have an anomalously long lifetime. Comparisons of the data with the theoretical calculations have been difficult, however, as some sets of calculated decay rates of $\pi^4\text{He}^+$ states have differed from each other by 1–2 orders of magnitude [2, 4, 5]. The transitions between short-lived states with a small principal quantum number $n_i$ for singly charged, two-body pionic helium ($\pi^4\text{He}^{2+} \equiv \pi^- + {}^4\text{He}^{2+}$) ions have been measured by X-ray fluorescence spectroscopy with a relative precision of approximately $2 \times 10^{-4}$ [41–44]. The atomic lines of $\pi^4\text{He}^+$ were not detected until very recently [9].

## 26.2 Experimental method

In the recent PSI experiment, laser pulses excited a transition from a pionic state of the neutral atom that had a nanosecond-scale lifetime, to a state with a picosecond-scale lifetime against Auger decay [5] (Figure 26.1). A $\pi^4\text{He}^{2+}$ ion was formed after Auger emission of the 1s

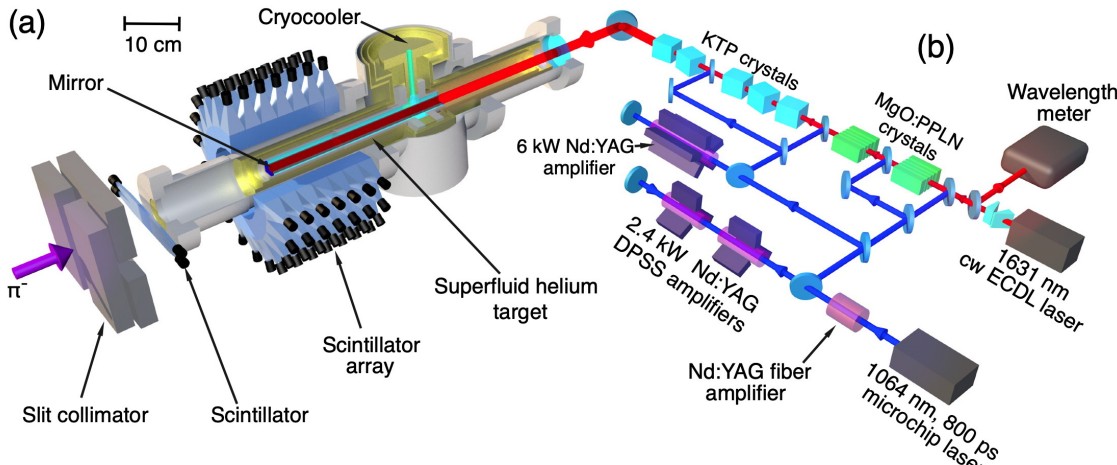

Figure 26.2: (a): Schematic showing the layout of the target used in the experiment. The $\pi^-$ beam passed through a scintillation counter and then came to rest in the cryogenic helium target. The resulting atoms are irradiated with $\Delta t = 800$ ps long laser pulses with wavelength $\lambda \approx 1631$ nm. (b): Schematic layout of the laser system, see text. From [9].

electron. Collisions with other helium atoms caused Stark mixing between the Rydberg and low $\ell$ orbitals of the ion [43, 45] as well as other possible effects [46]. This Stark mixing led to the absorption of the $\pi^-$ by the nucleus. The resonance condition between the laser beam and the $\pi^4\mathrm{He}^+$ atom was detected as a peak in the rates of neutrons, protons, and deuterons. This peak was superimposed on a background containing other $\pi^4\mathrm{He}^+$ atoms that decayed spontaneously with a lifetime of around $\approx 7$ ns [5, 37].

This experiment used the $\pi$E5 beamline [47] that provided a $\pi^-$ beam that had a momentum between 83 and 87 MeV/c, and an average intensity of $N_\pi = (2-3) \times 10^7$ s$^{-1}$. A Wien filter was placed upstream of the target. This filter diverted most of the contaminant $e^-$ that arrived at a rate $> 3 \times 10^9$ s$^{-1}$ into the blades of a slit collimator made of steel. The purified $\pi^-$ beam was focused into an elliptical beam spot that had a full-width-at-half-maximum (FWHM) horizontal size of 23 mm and vertical size of 15 mm. For this a pair of quadrupole magnets provided by the CERN magnet group was used. The $\pi^-$ beam passed through a plastic scintillator plate that had a thickness $t_d = 4.7$ mm. The plate was segmented into four sections with each section having a size of $20 \times 20$ mm$^2$. The beam then entered the experimental target.

The correlations between the arrival times $t_a$ and energy depositions $\Delta E$ of hits that occurred in the scintillator plates at the entrance of the target are shown in the contour plot of Figure 26.3 (a). The $\pi^-$ arrived in bursts spaced by regular intervals $\Delta t = 19.75$ ns. This arose from the $f_a = 50.63$ MHz radiofrequency of the 590 MeV cyclotron, with each RF cycle containing on average $N_\pi / f_a \approx 0.4 - 0.6 \ \pi^-$. The $\pi^-$ arrival events which are located in the rectangular area indicated by broken lines were distinguished from $\mu^-$ and $e^-$ in the beam by the time-of-flight methods and the estimated $\Delta E$ value of 2.6 MeV for $\pi^-$ in the scintillator plate.

Based on past experiments [37] we assumed that a 2.3% fraction of the $\pi^-$ that were able to come to rest in the superfluid helium target (Figure 26.2 (a)) with a length of 150 mm, diameter of 42 mm, and a temperature of $T = 1.7$ K formed the metastable variant of the atoms. A laser beam that had a diameter of $d = 25$ mm, a pulse length of $\Delta t = 800$ ps, pulse energy $E = 10$ mJ, repetition rate $f_r = 80.1$ Hz and wavelength $\lambda \approx 1631$ nm entered the target. The beam irradiated $> 60\%$ of the $\pi^4\mathrm{He}^+$ produced in the target. The implied

production rate of the pionic atoms of $> 3 \times 10^5$ s$^{-1}$ ensured that we retained a probability of coincidence of around $10^{-3}$ for a laser pulse to irradiate a $\pi^4$He$^+$ atom.

The nuclear fragments that emerged from the absorption of $\pi^-$ tended to follow tranjectories that were anticollinear [5,48,49] with a typical kinetic energy of a few tens of MeV. The arrival times $t_a$ and the energy depositions $\Delta E$ of the fragments were measured (Figure 26.3 (b)) by an array containing 140 plastic scintillation counters with size $40 \times 35 \times 34$ mm$^3$. These counters covered a solid angle of $\approx 2\pi$ steradians seen from the target. The size of the scintillation counters was chosen so that the detection efficiency for $E \geq 25$ MeV neutrons was significant ($< 10\%$) [5] while simultaneously achieving the discrimination condition which rejected most of the background $e^-$ from either $\mu^-$ decay or the particle beam. The background $e^-$ deposited an average energy $\Delta E = 6 - 8$ MeV. Monte Carlo simulations indicated that most of these events could be removed by rejecting those events an energy deposition of $\Delta E < 20 - 25$ MeV. The waveform [50–52] of the signal from the counters were recorded during each laser pulse arrival by using waveform digitizers that had sampling rates of $f = 3.06$ Gs·s$^{-1}$. We did this by developing a custom readout system, which used the DRS4 chip which is an application-specific integrated circuit (ASIC) that was based on switched capacitor arrays [53,54]. An earlier version of the electronics based on the DRS4 ASIC was used in an experiment to determine upper limits on the annihilation cross sections of antiprotons of kinetic energy $E \approx 125$ keV on thin target foils [51,55,56], the results of which were compared with the cross sections measured at higher energies $E = 5.3$ MeV [57,58].

Figure 26.3 (b) shows a $t_a - \Delta E$ contour plot of hits on the scintillator array surrounding the target. We selected those events that were within the area indicated by the broken lines. This removed most of the background $e^-$ as well as fission products with low velocities. The blue time spectrum of Figure 26.3 (c) shows the distribution of scintillator hits that were measured without any laser beam irradiating the atoms. The consecutive $\pi^-$ arrivals at $t = 0$ and at $t = 19.75$ ns produced a pair of peaks in the spectrum that contained the $> 97\%$ majority of $\pi^-$ that underwent nuclear absorption immediately after arriving in the target. The fraction $(2.1\pm0.7)\%$ that remained constituted a spectrum with a decay lifetime of $\tau = (7\pm2)$ ns in the intervals between the arrivals of $\pi^-$. This approximately agreed with the results of a Monte Carlo simulation [5] of the expected signal, and with an experiment carried out previously [37] using a target filled with liquid helium.

The laser pulses that reached the experimental target at a time $t = 9$ ns after the arrival of $\pi^-$ had a timing jitter of typically $\Delta t \leq 1$ ns. These laser pulses were produced by an injection-seeded, optical parameteric generator (indicated as OPG in Figure 26.2(b)) and amplifier (OPA) laser system. We constructed a diode-pumped solid state (DPSS) neodymium-doped yttrium aluminium garnet (Nd:YAG) laser that was of single pass design. The laser was precisely fired in synchronization with the RF of the cyclotron to pump the OPG-OPA laser. We based the OPG-OPA laser system on a continuous-wave (cw) external-cavity diode laser (ECDL) with a wavelength $\lambda \approx 1631$ nm. This seed beam was amplified using magnesium oxide doped periodically-polled lithium niobate (MgO:PPLN) crystals. This produced laser pulses of energy $E = 70$ uJ. OPA to $E = 10$ mJ was carried out in five potassium titanyl phosphate (KTP) crystals. The linewidth of the portion of the laser beam having a narrow spectral component was of order 10 GHz. These OPG and OPA processes introduced a 3 GHz uncertainty in the determination of the optical frequency of the laser pulses.

## 26.3 Experimental results

The experiments began by searching for the $(n, l) = (16, 15) \rightarrow (17, 14)$ transition by scanning a laser based on dye and Ti:Sapphire [59] pulse amplification over a 200 GHz wide region around the transition frequency $\nu_{th} = 781052.6(2.0)$ GHz which was calculated by theory [5]. The 2.0 GHz uncertainty is caused in large part by the experimental uncertainty on the mass

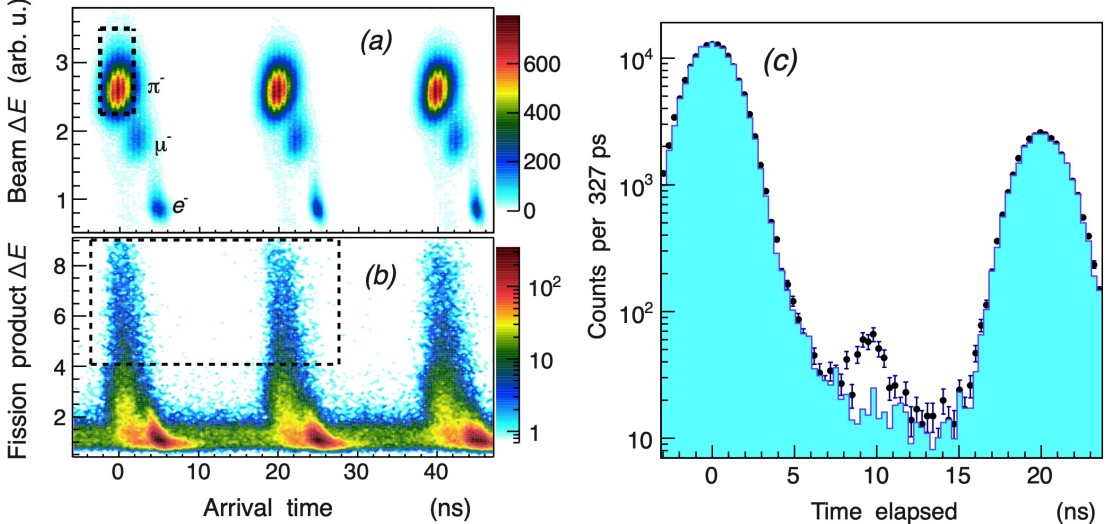

Figure 26.3: (a): A contour plot which shows the correlation between the arrival times $t_a$ and the energy depositions $\Delta E$ of particles that were measured by a scintillation counter placed at the entrance of the helium target. The type of particle was identified. The $\pi^-$ events in the rectangular region shown using broken lines were selected. (b): The $t_a - \Delta E$ plot of showing fission fragments that strike the scintillator array following $\pi^-$ absorption by the helium nuclei. Background $e^-$ with an energy deposition of $\Delta E < 20 - 25$ MeV were removed by accepting only the events in the region indicated by the rectangle. (c): The time spectra of nuclear fragments measured with (indicated by filled circles with error bars) and without (blue filled histogram) the laser irradiation at $t = 9$ ns. The peak in the former spectrum at $t = 9$ ns here corresponds to the laser resonance signal of $(17,16) \rightarrow (17,15)$. From [9].

of $\pi^-$. No significant signal was observed. The coupling of the resonance daughter state $(n, \ell) = (17, 14)$ to an electronically excited state of $\pi^4\text{He}^+$ is theoretically expected to cause large scalar and tensor polarizabilities of amplitudes $4 \times 10^4$ and 70 atomic units, respectively [6], and this is believed to destabilize the daughter state against atomic collisions [60, 61].

We next searched for the $(16, 15) \rightarrow (16, 14)$ resonance at a theoretical transition wavelength $\lambda = 1515.3$ nm. The 250 fs lifetime [5] of the daughter state $(16, 14)$ should give rise to a large resonance width $\Gamma_A = 640$ GHz. Experimental data that corresponded to $> 6 \times 10^7$ detected $\pi^-$ arrivals showed no signal that was statistically significant. The reason why the resonance was not observed is not understood. One possibility is that collisions with other helium atoms may destroy the $\pi^-$ population that occupies the parent state $(n, \ell) = (16, 15)$. Similar effects have been observed in several states of $\overline{p}\text{He}^+$ atoms [62–65]. Alternatively, it may be that only a negligible fraction of $\pi^-$ are captured into state $(n, \ell) = (16, 15)$, as has been observed for some states of lower $n$ in the $\overline{p}\text{He}^+$ case [66–69].

We searched for the transition $(17, 16) \rightarrow (17, 15)$. The time spectrum indicated by filled circles in Figure 26.3 (c) was measured by accumulating data from $2.5 \times 10^7$ $\pi^-$ arrivals with the laser wavelength tuned to $\lambda \approx 1631.4$ nm. A peak was observed at $t \approx 9$ ns which contained some 300 events. The signal-to-noise ratio was 4 and the statistical significance $> 7$ standard deviations. Its width $\Delta t = 2$ ns was compatible with the expected dispersion of the time-of-flights of the fission fragments that arrive at the scintillator array. We found that the rate of 3 h$^{-1}$ of detected resonant $\pi^4\text{He}^+$ events is roughly compatible with the production rate of $> 3 \times 10^5$ s$^{-1}$ of the atoms and with Monte Carlo simulations [5] that were carried out by assuming that most of the metastable population are captured into the parent state

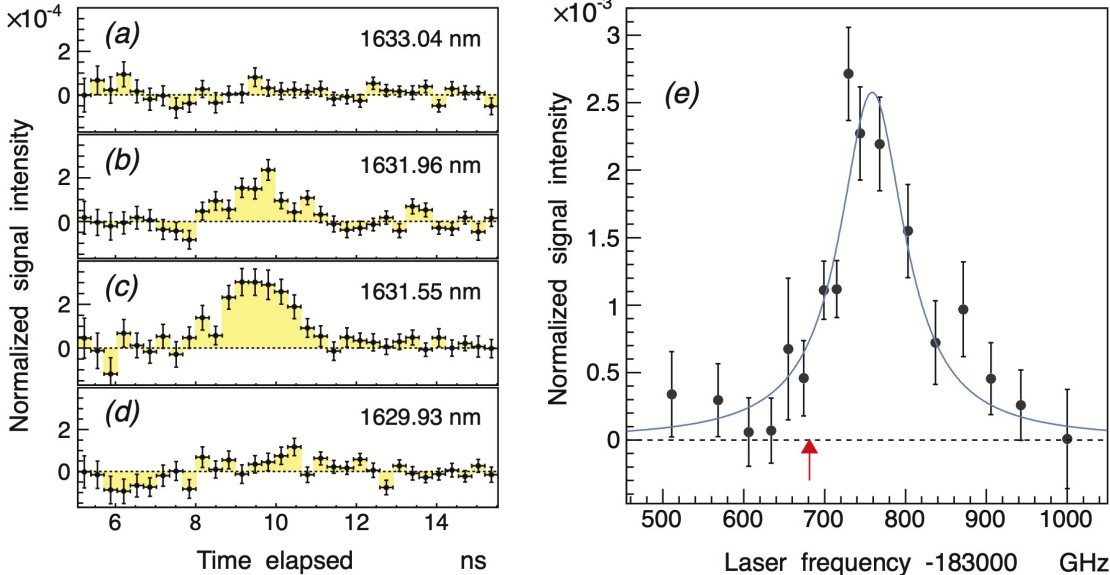

Figure 26.4: (a)–(d): The normalized time spectra of the resonance signal of the $\pi^4\mathrm{He}^+$ transition $(n,l) = (17,16) \rightarrow (17,15)$ which was measured at four laser wavelengths. The spectra were obtained by taking the difference between the timing distributions of $\pi^-$ absorption that were measured with and without the laser irradiation. (e): The profile of the resonance measured by scanning the laser frequency over a 500 GHz wide region. The red arrow indicates the position of the spin-averaged transition frequency obtained by a three-body QED calculation [5]. From [9].

$(n,\ell) = (17,16)$. When the laser was detuned off the resonance frequency (Figure 26.4 (a)-(d)), the signal proceeded to decrease and disappear.

The resonance signal intensity (Figure 26.4(a)–(d)) was obtained by taking the difference between the normalized time spectra that were measured with and without laser irradiation. The number of detected events under the induced peak around $t = 9$ ns was then counted. The resonance profile of Figure 26.4(e) was obtained by scanning the laser frequency. Each data point shown here contains data that were collected over a 20–30 h period of the experiment. The statistical uncertainty that arises from the finite number of $\pi^4\mathrm{He}^+$ events is indicated by vertical error bars. The measured width of $\approx 100$ GHz of this resonance agrees with a convolution of the expected 33 GHz Auger width [5] of the daughter state $(n,\ell) = (17,15)$ calculated by theory, collisional and power broadening [7] which are estimated to cause a contribution of $\approx 50$ GHz, and the $\approx 10$ GHz linewidth of the narrowband spectral component of the laser pulses. Some further broadening of this resonance may be caused by atomic collisions that shorten [6, 62] the lifetime of the resonance daughter state $(n,\ell) = (17,15)$. The spacing of 3.0 GHz [5, 70] between the fine structure sublines that is expected from the interaction between the electron spin and the orbital angular momentum of $\pi^-$ cannot be resolved in our experiment since it is much smaller than the 33 GHz natural width of the resonance itself. The best fit (see blue curve) of two overlapping Lorentzian functions which take these sublines into account was shown to have a reduced $\chi^2$ value of 1.0. The resonance centroid is $\nu_{\mathrm{exp}} = 183760(6)(6)$ GHz. The statistical uncertainty of 6 GHz is due to the finite number of detected $\pi^4\mathrm{He}^+$. The systematic uncertainty of 6 GHz contains the contribution of 5 GHz that is related to the selection of this fit function as well as other contributions related to the laser.

This $\nu_{\text{exp}}$ value determined in the experiment is larger by $\Delta \nu = (78 \pm 8)$ GHz compared to the theoretical value [5] $\nu_{\text{th}} = (183681.8 \pm 0.5)$ GHz. This shift in the resonance frequency is believed to be caused by collisions with other helium atoms [7]. Some similar effects have been previously observed [62,71] for some $\overline{p}\text{He}^+$ resonances. The gradient of this shift that is expected at a target temperature $T = 4$ K was calculated to be $d\nu/d\rho = (4.4 - 6.5) \times 10^{-21}$ GHz·cm$^3$ using the impact approximation of the binary collision theory of spectral lineshapes [7]. At the density of the superfluid target used in these experiments, the blueshift expected from theory corresponds to between $\Delta \nu = 96$ and $142$ GHz. This theoretical result roughly agrees with the result of the experiment. This collisional shift must be experimentally measured before the $\pi^-$ mass can be determined.

In future experiments, we are planning to search for other transitions such as $(n,l) = (17,16) \rightarrow (16,15)$ that should be narrower by a factor of at least $10^{-3}$ compared to the recently-detected transition using helium gas targets where the collisional shifts are small. Laser spectroscopic techniques that enable higher precision are available [22–24, 59]. The precision of the calculated transition frequencies $\nu_{\text{th}}$ is now limited by the experimental uncertainty of the $\pi^-$ mass, but the precision of the calculations themselves [5] can be improved to a fractional precision of less than $10^{-8}$ for some transitions as in the HD$^+$ [72, 73] and $\overline{p}\text{He}^+$ [19,20] cases. These pionic experiments at PSI will also complement the measurements on $\overline{p}\text{He}^+$ that will be carried out at the ELENA facility [74–76].

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
