# Peer review of "Recent results of laser spectroscopy experiments of pionic helium atoms at PSI"

_SciPost Physics, doi:SciPost Phys. Proc. 5, 026 (2021)_

## Round 1 · Referee Report · Adrian Signer · 2021-1-14

Report
We (the editors Cy Hoffman, Klaus Kirch, Adrian Signer) had the
opportunity to review an earlier draft of the article and were in
communication with the authors before the submission. All our
comments and suggestions have been taken into account. Hence, we
think the paper can now be published in the current form.
Anonymous on 2021-02-10 [id 1224]
We thank Anonymous referee 2 for useful comments which we will address.
>The highest published precision for pi4He is DE =+- 2 eV [37], i.e. one should >change … to about 10-4 -> … up to 2 x 10-5.
If a X-ray transition energy of 10.7 keV were measured with a precision of 2 eV, the resulting relative precision would seem to be 2x10^-4. Not 2x10^-5. To achieve a relative precision of 2x10^-5 on 10.7 keV, one would need a precision of 0.2 eV which was never achieved as far as we know on pionic helium.

---

## Round 1 · Referee Report · Anonymous · 2021-2-9

Report
• Report on manuscript
Recent results of laser spectroscopy experiments of pionic helium atoms at PSI
by M. Hori et al.
for Review of Particle Physics at PSI, doi:10.21468/SciPostPhysProc.2
The establishment of the laser-induced decay of the pi4He+ system is a significant step in experimental performance due to the continuous developments both in laser technology as well as in beam quality provided for pions at PSI. It confirms directly the supposed existence of high-lying metastable states also for mesic exotic atoms. The experiment constitutes the first step of continuation with pions of the successful experiments with antiprotonic helium, which lead to an impressive increase of precision both in atomic theoretical and experimental methods.
As the results are given in a recent Nature article which addresses the details of the experiment and to which the wording of the presented manuscript is similar, and in view of the scope of the review to give an overview on PSI physics I suggest a slightly different introduction strategy for the presentation in this journal.
In summary:
• The paper and its topic are important to be published in the compilation of PSI physics.
• Some changes of the text are suggested (details below).
Requested changes
- Original line numbers are kept where changes are suggested.
- Comments are given inside >>>> ....<<<<.
- I apologize for the bad quality of original text directly copied from the pdf.
1
Introduction
The introduction suggests the description of a planned experiment and its perspectives of a proposed. There is, however, already a substantial result achieved. A rearrangement of the introduction in about the following manner would provide a better guiding for non-experts.
20 Metastable pionic helium is a neutral exotic atom [1–4] that contains a helium nucleus with
21 an electron in the ground state, and a negatively-charged pion (_ ) occupying a state having
22 high principal and orbital angular momentum quantum numbers of around n _l+ 1 _ 16.
23 These states have nanosecond-scale lifetimes against the competing cascade processes of _
24 nuclear absorption and _ ! _ + __ decay. This longevity arises because the _ orbitals
25 have very small overlap with the nucleus, and so the rates of electromagnetic cascade pro
26 cesses involving the rapid deexcitation of the _, such as Auger and radiative decays are
27 significantly reduced.
The transition energies
44 between short-lived states with a small principal quantum number ni for singly charged, two-
45 body pionic helium (_4He2+ _ _ + 4He2+) ions have been measured by X-ray fluorescence
46 spectroscopy with a precision up to 2 x 10-5 in the 1970’s or earlier [37–39]. The atomic lines of
47 _4He+ were not detected until very recently [5].
>>>>
Comments:
The highest published precision for pi4He is DE =+- 2 eV [37], i.e. one should change … to about 10-4 -> … up to 2 x 10-5.
There is another pi4He X-ray experiment from the 80ies: S.Baird et al., Nucl.Phys.A 392 (1983) 297.
<<<<
38 The existence of long-living piHe+ atoms has been inferred in an indirect way from four experiments
39 [29–33] that were initially carried out using early synchrocyclotron facilities [34,35] and liquid
40 helium bubble chambers [36]. All these experiments observed that some _ coming to rest in
41 helium targets have an anomalously long lifetime. Comparisons of the data with the theoretical
42 calculations have been difficult, however, as some sets of calculated decay rates of pi4He+
43 states have differed from each other by 1–2 orders of magnitude [2, 4, 6].
27 This allows to carry out laser spectroscopy [5, 6] of pi4He. By comparing the atomic frequencies measured by laser spectroscopy with the re-
31sults of quantum electrodynamics (QED) calculations, the _ mass [7–9] can, in principle, be
32 determined with a high precision. This can help set improved upper limits on constraints on the muon
33 antineutrino mass by laboratory experiments [10]. Some upper limits may also be set on any
34 exotic force [11–15] that involves the _ , as has been done in the case of antiprotonic helium
35 (pHe+ _ p+He2+ +e ) atoms [16–26]. Unlike the pHe+ case, the atomic structure of _4He+
36 contains no hyperfine structure that arises from the spin-spin interaction between the spin-0
_37 and 4He nucleus [27,28].
>>> Comment: The level of precision achievable by the QED calculations (<= 10-8) – without the uncertainty stemming from the pion mass - should be stated.
<<<<
2
Experimental method
51 Auger decay [6] (Figure 26.1). A This two-body pi4He2+ ion was formed after Auger emission of
52 the 1s electron. The ion was then promptly destroyed by collisions with other helium atoms,
53 which caused Stark mixing between the Rydberg and low ` orbitals of the ion [38, 40] as
well as neutralization by electron transfer other possible effects [41]. This Stark mixing …
>>>>
Comment: The original sentence line 51 sounds somehow disconnected to the intended meaning?
<<<<
56 as a peak in the rates of neutrons, protons, and deuterons originating from pion absorption.
57 a background containing other metastable pi4He+ atoms that decayed …
76 We assumed that a 2.3% fraction of the _that were able to come to rest in the superfluid
77 helium target (Figure 26.2 (a)) with a length of 150 mm, diameter of 42 mm, and a temper-
78 ature of T = 1.7 K formed the metastable variant of the atoms [33]. …
>>>>
Comment: To help the reader, one should point out that the assumption for the fraction 2.3% is based on the (rather precise) experimental result ((2.30 +- 0.07)%) of [33]
<<<<
>>>>
Comment: Fig 26.2 caption lines 4/5, line 79/80 is an almost one-to-one repetition (all information needed twice?)
<<<<
89 size of the scintillation counters was chosen so that the detection efficiency for E _ 25 MeV
90 neutrons was significant (< 10%) [6] while simultaneously achieving ...
>>>>
Comment: The efficiency value (<10%) is confusing in connection to En >=25 MeV. Better to write <=10%.
>>>>
>>>>
Comment: Though it summarizes presumably a big technical effort, the details lines 96-101 may be too specific in the context of this article. Perhaps one should consider to omit them.
<<<<
3
26.3 Experimental results
>>>>
Comment: Start new paragraph with line 135.
135 We next searched for the (16, 15) - (16, 14) resonance. …
It would be instructive to give the theoretical frequency also for this transition.
<<<<
>>>>
Fig. 26.4(e): Caption - the meaning of the red arrow (theoretical value) should be stated explicitly e.g. by a cross reference at line 177.
<<<<
4
26.4 OUTLOOK
>>>>
The very interesting perspectives following from this success - the very last part - the physics and technical challenges in order to exploit this discovery deserve to my opinion an extra subchapter 26.4 (entitled OUTLOOK or PERSPECTIVES etc.).
One should explain here in a little bit more clear way the roadmap from dnu/nu about 10-6 (Large Auger widths) to dnu/nu about 10-8 (negligible Auger width).
So, this subchapter should start with the first problem, the precision achievable for the pion mass in view of collisional shift (addressed in the last sentence of the one but last paragraph lines 184/185: This collisional shift … can be determined.).
Secondly, the search for narrow transitions (line 187: narrower by a factor of at least 10-3 -> 10+3) together with a statement on the laser band width to achieve a good determination of the experimental frequency should give the reader a feeling of the technical expertise.
<<<<

---

## Round 2 · Referee Report · Anonymous (Referee 2) · 2021-3-6

Report

The changes made for V2 are sufficient.

I have only 3 marginal editorial remarks (see requested changes).

Requested changes

  1. Caption Fig. 26.2. The sentence starting - This resulting atoms ... seems to be grammatically incorrect.

2. line 65: 20 x 20 mm -> 20 x 20 mm^2

3. line 109: 7+-2 ns -> (7+-2) ns

---

## Round 2 · Author Response

We would like to thank the referees for giving us such useful comments. This is our resubmission.

---

## Round 2 · List of Changes

We would like to thank the referees for giving us such useful comments. This is our resubmission.

  1. "The highest published precision for pi4He is DE =+- 2 eV [37], i.e. one should change … to about 10-4 -> … up to 2 x 10-5."

As 2 eV/10 keV = 2x10^-4 and not 2x10^-5, we modified it to read 2x10^-4.

  1. "There is another pi4He X-ray experiment from the 80ies: S.Baird et al., Nucl.Phys.A 392 (1983) 297."

The reference was added.

  1. "The level of precision achievable by the QED calculations (<= 10-8) – without the uncertainty stemming from the pion mass - should be stated."

This is included in the last paragraph of the text.

  1. "The original sentence line 51 sounds somehow disconnected to the intended meaning?"

We modified it to read, "A two-body...."

  1. "To help the reader, one should point out that the assumption for the fraction 2.3% is based on the (rather precise) experimental result ((2.30 +- 0.07)%) of [33]"

The 2.3% value is somewhat arbitrary in the sense that it depends on where one cuts on the time spectrum to define the long-lived fraction and the distribution of the pions that come to rest in the experimental target. Since our experimental setup and beam momenta are very different from those of Ref. [33], the 2.3% value is taken to be an approximation.

  1. "Fig 26.2 caption lines 4/5, line 79/80 is an almost one-to-one repetition (all information needed twice?)"

The information was removed.

  1. "The efficiency value (<10%) is confusing in connection to En >=25 MeV. Better to write <=10%."

The efficiency is smaller than 10%. A more precise upper limit cannot be reliably estimated based on the known efficiencies of these detectors from the literature.

8."Though it summarizes presumably a big technical effort, the details lines 96-101 may be too specific in the context of this article. Perhaps one should consider to omit them."

The technical efforts described here corresponds to the work of PSI staff members and so out of respect to PSI we would like to highlight them.

  1. "Start new paragraph with line 135. It would be instructive to give the theoretical frequency also for this transition."

We added the wavelength to the sentence.

  1. "Fig. 26.4(e): Caption - the meaning of the red arrow (theoretical value) should be stated explicitly e.g. by a cross reference at line 177."

We added the sentence, "The red arrow indicates the position of the spin-averaged transition frequency obtained by a three-body QED calculation"

11. "The very interesting perspectives following from this success - the very last part - the physics and technical challenges in order to exploit this discovery deserve to my opinion an extra subchapter 26.4 (entitled OUTLOOK or PERSPECTIVES etc.)." "Secondly, the search for narrow transitions (line 187: narrower by a factor of at least 10-3 -> 10+3) together with a statement on the laser band width to achieve a good determination of the experimental frequency should give the reader a feeling of the technical expertise. "

This proceedings paper is intended to summarize past work. Future possibilities and an estimation of the achievable precision would need a more through treatment which is not within the scope of the present paper. We added the sentence, "Laser spectroscopic techniques that are commensurate with a higher level of precision are available. "

---

## Editorial Decision

published